# Design as an Indicator of Tourist Destination Change: The Concept Renewal Cycle at Watkins Glen State Park

Hans Klein-Hewett 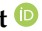

Department of Landscape Architecture, Iowa State University, Ames, IA 50011, USA; hansk@iastate.edu

**Abstract:** For decades, the Tourism Area Life Cycle (TALC) model, its iterations, and its critics have shaped the conversation about change and adaptation at tourist destinations. However, few life cycle models consider the designed landscape as a factor in the evolutionary process or as a signifier of change. This oversight is problematic because the landscape, the aggregation of consciously designed spaces and amenities, is where tourism takes place. It is the physical manifestation of the tourist destination and therefore significantly influences how the site is organized, consumed, and evaluated. To illustrate the landscape's importance, this article proposes a new life cycle model called the Concept Renewal Cycle (CRC), which tracks the intent of the designed landscape, the concept, to understand and document destination change. The model introduces and utilizes relevancy as the variable that determines concept success and instigates action. The proposed model and other prominent life cycle models are analyzed and compared through the case study of Watkins Glen State Park in New York state. While the other models struggle to reflect the evolution at Watkins Glen, the CRC shows resilience by eschewing TALC's inevitable, time-based decline structure in favor of a cyclical pattern where concept revision allows for prolonged maturity.

**Keywords:** life cycle; concept revision; landscape tourism; landscape architecture; scenic destinations; design decisions; conceptual design

## 1. Introduction

Economists and geographers organize shifts in amenities, infrastructure, tourists, and financial capital into theoretical development patterns called life cycle models. These models analyze how and why tourist destinations evolve and, ideally, allow researchers and tourist managers to anticipate future change. Many models utilize visitation and carrying capacity—the balance between physical space and social comfort—to signify when infrastructure changes are required.

However, most life cycle models do not consider the arrangement or design of the landscape amenities when tourist destinations change [1]. When viewed as a static object, the consciously designed landscape (i.e., programmed spaces, site amenities, structures, plant material, ornamentation) is a collection of material artifacts: symptoms and constructs representing how destination managers historically understood their clientele, their tourism product, and the tourism industry [2]. Therefore, the landscape is the document that chronicles why a change was necessary and how managers addressed the issue. Simultaneously, the landscape is the vehicle for instituting change, so the landscape is an active participant in shaping how tourists perceive and consume the destination. Recognizing this dual nature of the designed landscape is essential when understanding and documenting how destinations change.

To rectify this gap, this article proposes a new theoretical life cycle model called the Concept Renewal Cycle that utilizes the landscape design concept and its relevancy to managers' preferred clientele as the variable that instigates change. The proposed model and other prominent life cycle models are then compared by applying a case study at Watkins Glen State Park, a well-known public scenic destination located in the Finger Lakes Region of New York state.

*Research Goals*

This research aims to propose and test a design-focused tourism life cycle model that documents change without using the conventional economic success metrics of visitation and carrying capacity. Instead, the design concept and its manifestation in the landscape becomes the variables that chronicle how destinations adapt to change. Using this approach, the landscape becomes the medium that can be proactively manipulated to change the design concept and achieve long-term destination maturity. Additionally, this research aims to illuminate the importance of design in tourism research, expand the scope of tourism studies, and foster collaboration between design and tourism scholars.

## 2. Materials and Methods

The Concept Renewal Cycle model proposed in this study was developed by analyzing how and why landscape architects and other designers approach design challenges. In educational and professional settings, landscape architects create design concepts that attempt to unify the designed materials, forms, and spaces into an identifiable and cohesive experience. When the landscape becomes outdated, underutilized, overused, or otherwise needs to be refreshed, designers are employed again to develop a new concept for the site, allowing the landscape to take on new meaning and form, thereby prolonging the site's longevity. The Concept Renewal Cycle attempts to apply the common design practice of concept creation and revision to tourism spaces to illustrate how the designed landscape can accommodate long-term destination maturity.

The proposed model and other prominent life cycle models are compared and tested in this study through a case study at Watkins Glen State Park. Located in the Finger Lakes Region in central New York state, Watkins Glen has been a famous scenic destination since 1863. The park has outlived seismic developments in transportation, shifts in socioeconomic classes and associated tourism consumption, and ownership changes. In 2015, it was named the third best state park in the United States, according to a poll from USA Today [3]. Despite only being open annually between May and November, the park welcomed over a million visitors for the first time in 2019 [4]. Its popularity is due to its complex role in the tourism marketplace; it is a destination for sublime scenery, geologic and cultural history, and outdoor recreation. Due to its popularity and nearly 160-year history, Watkins Glen is an ideal case study to understand the role that design plays as tourist destinations evolve.

The case study at Watkins Glen is organized through a longitudinal review of major design changes from the park's opening in the 1860s to the latest significant design change in 2018. To establish that history, primary source materials were gathered from the archives at Cornell University and Schuyler County Historical Society. As the park started as a private destination, few public records describe how or why the park changed between 1863 and 1906. Therefore, the primary sources of history came from Watkins Glen guidebooks, advertisements for the park, and local and regional newspaper articles published between 1860 and 1910. To better understand the park's significance and reach, national newspaper and magazine articles were analyzed to document who was discussing the park and from where they came. After the park became publicly owned in 1906, institutional reports provided most of the information on infrastructure changes and why the improvements were needed. This information was supplemented by local newspaper articles which documented the day-to-day activities and described significant events. To document aesthetic changes, postcards, stereographs, and historical photographs were analyzed to confirm design style and ornamentation.

The landscape design changes are then organized further through an exploratory process that seeks to find patterns and relationships in amenities and aesthetic styles [5]. Specifically, this study organizes the design changes into eras primarily based around common aesthetic styles or obvious ways the site was meant to be experienced. This information is supported by broader contextual research acquired through literature review and broad cultural and historical analysis of tourism trends. Finally, the case study and its design eras are employed to compare prominent existing life cycle models to the proposed

model through analytical generalization [6]. This process aims to illustrate the differences between the existing and proposed models and highlight areas for further study.

## 3. Review of Literature

### 3.1. Review of Tourism Destination and Tourism Concept Literature

In abstract terms, the tourism industry develops, markets, and sells commodities commonly called tourism products. A tourism product is the combination of attractions, services, amenities, ideas, and experiences that are continually produced and sold to tourists. While scholars have dissected tourism products in numerous ways [3–5], this article will utilize Smith's model, in which the core of the tourism product is the "physical plant"—the physical source of the tourism activity, such as a beach, historic district, or, in Watkins Glen's case, a rocky gorge. The physical plant is reliant on its services (e.g., additional amenities, management, maintenance, salespeople) and hospitality (e.g., service quality) to make the destination accessible to tourists [7,8].

When the physical plant and its supporting amenities, facilities, activities, interests, and attractions are combined, the resulting geographic space is called a tourist destination [9]. A destination works on multiple geographic levels (e.g., location of the physical plant, its vicinity, its region), so it is often not defined by specific geographic boundaries. A destination is a static object because it is a known place and singular marketed product. It is also a dynamic process because it includes an ever-changing mix of agents, products, tourists, and demands [10].

The destination's identity—how it is understood and perceived in the mind of tourists, locals, and tourism managers—is a social construct called the concept of the destination [11]. The concept is a discourse between the idea of the destination (i.e., knowledge about and meaning of the destination, formed by maps, signs, guidebooks, amenities, and media) and the actions that produce or reinforce that idea (i.e., infrastructural improvements in quality and quantity) [12]. As a concept is both static and dynamic, it produces a continually changing outcome. That outcome—the destination's concept—is what tourists use to identify and evaluate the destination [11].

### 3.2. Review of Tourism Life Cycle Literature

Over the last forty years, the topic of tourist destination change has been heavily influenced by Richard Butler's Tourism Area Life Cycle (TALC) model [13]. In Butler's model, a destination goes through six development stages: "exploration" (a small number of tourists verify tourism interest), "involvement" (more tourists; locals respond to accommodate visitors), "development" (external developers promote continued growth), "consolidation" (rapid rise in visitors, then the beginning of a slowdown), "stagnation" (peak visitor capacity, economic stasis), and finally "decline" or "rejuvenation" (tourism venture fails or the source of attraction is changed to attract new clientele) (see Figure 1).

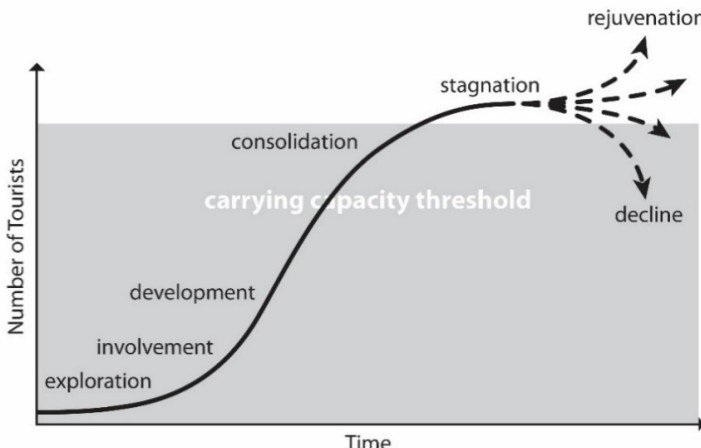

**Figure 1.** Tourist Area Life Cycle (TALC). Graphic by author, adapted from Butler, 1980.

While numerous scholars have found the TALC model helpful in providing a conceptual framework upon which economics, trends, and influences can be tracked, others, including Butler himself, have critiqued the model. A common critique involves the TALC's assumption of inevitable decline. Several researchers found case studies where the "decline/rejuvenation" phase was avoided due to a stable tourism market, the consolidation of competitive destinations, or the rejuvenation of only declining amenities [14–16]. These researchers call the prolonged stability at a tourist destination the "maturity" stage to demonstrate that decline is not inevitable. However, it must be noted that those studies did not consider the landscape or its designed amenities as signifiers or evidence of change. Another common critique is TALC's reliance on carrying capacity to determine life cycle change [17]. Carrying capacity is defined as "the maximum number of visitors which an area can sustain without unacceptable deterioration of the physical environment and without considerably diminishing user satisfaction" [18,19]. Scholars have had difficulty defining carrying capacity because it is based on a combination of psychosocial preferences and site infrastructure, variables that are unique to each user and each destination [14].

There are two pertinent life cycle models which incorporate infrastructure change as a mechanism for destination maturity. The first is Agarwal's adaptation of the TALC model, in which a "reorientation" stage is added after the TALC's "stagnation" stage (see Figure 2). In this new stage, managers evaluate and respond to the threat of decline by adjusting the infrastructure or amenities to meet demands. By making this adjustment and repeating the reorientation stage as necessary, the destination can adapt to external changes and extend its life [20].

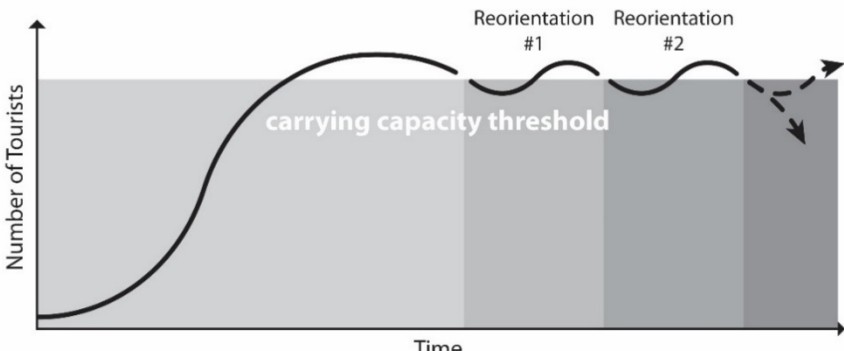

**Figure 2.** TALC model considering Agarwal's reorientation phases. Graphic by author, adapted from Agarwal, 2006.

The second pertinent model comes from Haraldsson and Ólafsdóttir, who took the "reorientation" idea further by applying Plog's psychographic visitor model [21]. Plog argues that users can be classified based on their travel patterns and preferred destinations. Plog places tourists on an "adventurous" to "non-adventurous" continuum, where "adventurous" tourists (Venturers or Allocentrics) seek less commercial and less refined destinations and "non-adventurous" tourists (Dependable or Psychocentrics) seek familiar and non-challenging destinations. He also creates a life cycle model where destinations transition from attracting adventurous tourists to attracting non-adventurous tourists by adapting facilities to increase visitation [21–23].

Haraldsson and Ólafsdóttir combine Plog's and Agarwal's models into a life cycle model based on expectations about nature purity and carrying capacity (see Figure 3). Instead of a continuum of "adventurous" to "non-adventurous" tourists, Haraldsson and Ólafsdóttir place tourists on a continuum of "strong purists" to "non-purists," based on the level of preferred amenities found at natural destinations. The first visitors to a natural destination, the "strong purists," seek a remote and natural experience. As the site is developed to accommodate more tourists, it is made less natural and more urban, which drives the "strong purists" to other, more remote destinations. Instead of the destination entering decline, a new group of tourists, seeking that newly offered, less rugged balance

of nature and urban, take their place. That process continues as additional infrastructure decreases the naturalness of the environment and, in turn, draws new users who are looking for their preferred balance [24].

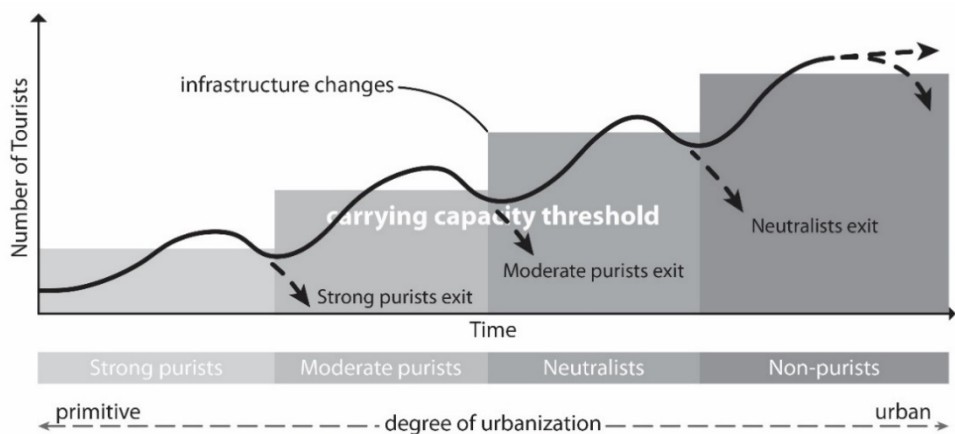

**Figure 3.** Adaptation of TALC considering Haraldsson and Ólafsdóttir's Purity Scale. Graphic by author, adapted from Haraldsson & Ólafsdóttir, 2018.

## 4. Proposal: The Concept Renewal Cycle

### 4.1. Relevance and the Concept Renewal Cycle

This article proposes an evolutionary, cyclical model that utilizes the concept of a destination and its conceptual iterations as indicators of change and the means to which destinations can prolong their maturity. Called the Concept Renewal Cycle (CRC), the model consists of three distinct phases: "concept creation," "concept maintenance," and "concept revision" (see Figure 4).

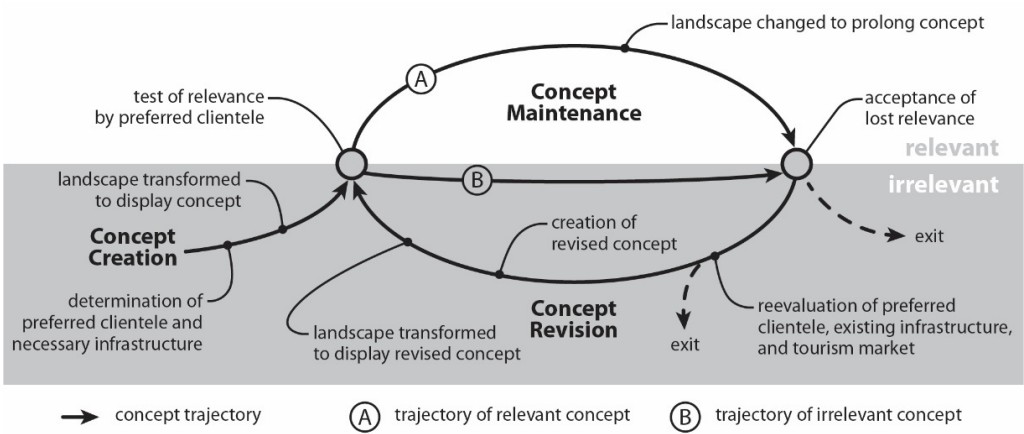

**Figure 4.** Concept Renewal Cycle Diagram. Source: Author.

This model uses the idea of "relevance" as the primary variable which instigates life cycle change. Simply defined, relevance is "the degree to which something is related or useful to what is happening or being talked about" [25]. In the tourism arena, the degree of relevance is different for managers and visitors. For managers, relevance is the degree to which the destination's conceptual identity meets or exceeds the users' demands and expectations. To determine the degree of relevance, managers must understand their destination, their tourism product, competitors' products, current and past trends, and their preferred clientele' demands and expectations. The primary concern for managers is whether the tourism product and its concept fit their preferred clientele's needs. If it does, their tourism product's relevance will resonate with their visitors, and the destination will

likely be successful. If it does not, their preferred clientele will seek other more relevant destinations. Managers will then need to either make the necessary revisions to their tourism product to attain/reattain relevancy or adjust their expectations of ideal clientele.

For the tourist, relevance is the degree to which the destination's conceptual identity meets their expectations and demands. To evaluate, the tourist, like the manager, must understand the tourism zeitgeist to determine their ideal destination. However, tourists must also consider the activities and influences of their socioeconomic peers because tourism is performative. Tourists use what Dean MacCannell calls "markers" (e.g., stories, photographs, writings, films, or other period-specific methods of destination consumption and dissemination) to evaluate destinations and their concepts as appropriate or not, based on this continually reproduced performance [26]. For a destination to be relevant, its concept and tourism product must meet tourists' needs and fit their peers' expectations.

### 4.2. The Concept Renewal Cycle Process

The first phase of the CRC is "concept creation," the formation and eventual manifestation of a destination's concept (i.e., its aesthetic, purpose, and identity). This phase happens when a new tourist destination is created, as the concept has not been defined or tested. To develop the concept, managers must consider the destination itself (i.e., what it can offer) and its context (i.e., what competitors are offering, what tourists want, where it could fill a void). Additionally, managers must identify the socioeconomic class of their preferred clientele. This identification is essential to determine the quality and quantity of amenities to offer at the destination. The "concept creation" phase ends once the infrastructure to attract the chosen clientele is constructed. At this point, the destination and its concept are available to tourists, and relevance is first tested.

The test of relevance is not a simple act for tourists to perform because it requires understanding their desires, the tourism market, and the appropriate behavior and tastes of others in their socioeconomic class [26]. However, users subconsciously perform this test daily as they determine what businesses to patronize, what to consume, and what clothes to wear, among a myriad of other decisions that determine and reinforce social behavior.

The second phase is "concept maintenance." In this phase, tourism managers attempt to prolong and reinforce the destination and its concept as a relevant option. As Figure 4 shows, there are two primary trajectories for the concept during the "concept maintenance" phase. The first trajectory (labeled as "A" in Figure 4) assumes the destination's concept is relevant to the preferred clientele. If so, the destination will likely see sustained or increased visitation as positive reviews and "markers" of the site are disseminated [26]. Managers might adjust the amenities to ensure the concept can weather small changes in tourist preferences, but the general concept remains the same.

The second trajectory (labeled as "B" in Figure 4) assumes the concept is not relevant to the preferred clientele. There is tension in this trajectory, as the managers' intentions did not resonate with their preferred clientele, though that does not foretell a decline. The term decline is intentionally not used in this model because it implies an economic trajectory. While destinations with relevant concepts are, in theory, more likely to be financially successful, economic success is not guaranteed. Similarly, destinations with irrelevant concepts may still be economically successful, as the presented concept may resonate with unintended tourists (i.e., tourists looking for that specific concept). More studies are warranted to determine the impact of the concept on economic success and the role of decline in determining conceptual changes.

The "concept maintenance" phase ends when managers accept that the proposed concept is no longer relevant. As the TALC suggests, relevance may be lost due to the number of tourists exceeding the physical or environmental carrying capacity. However, relevance may instead be lost due to socioeconomic changes, transportation shifts, technological advances, broad perception shifts, economic downturns, increased competition, budgetary constraints, or a multitude of other reasons. Regardless, the destination is forced to reconsider its concept and investigate options to reconnect with its preferred clientele.

The length of the "concept maintenance" phase, the time between the test for relevance and the acceptance of lost relevance, is intentionally left undefined. This is because the maintenance length is dependent on how long it takes managers to accept the loss of relevance. If a concept is relevant, the "concept maintenance" phase may span several decades, largely dependent on the reception to the concept and the manager's ability and desire to change. If the concept is deemed clearly irrelevant, managers may accept that a change is required shortly after the concept is manifested in the landscape and immediately begin to make changes and present a new concept.

The third phase in the CRC is "concept revision:" the reckoning and acceptance of a loss of relevance and the reconsideration of the concept to adapt to a new reality. Where the "concept maintenance" phase tries to fight change, "concept revision" accepts the change and starts to reconsider the destination's place in the tourism zeitgeist. If managers determine a relevant concept cannot be attained, managers may decide to exit the tourism marketplace. That decision can be made at any time during the revision process, as realities about the theoretical distance between the destination and tourist preferences are revealed. However, through the reevaluation of their preferred clientele, existing infrastructure, and tourism market, managers have the option to develop and present a new concept for their destination. Tourism managers then start the Concept Renewal Cycle again with a second "concept creation" phase, this time basing their destination's concept on existing users, programming, and infrastructure. Similarly, the "concept revision" phase may involve minor edits to the concept, which can be implemented quickly, or may require multi-year planning and construction projects.

## 5. Case Study: Landscape Design Changes at Watkins Glen

### 5.1. Introduction to Watkins Glen State Park

To test the Concept Renewal Cycle and compare it with other previously mentioned life cycle models, this article uses a case study at Watkins Glen State Park. Found in the Finger Lakes Region of New York State, the park is located at the southern end of Seneca Lake and is directly east of the Village of Watkins Glen (see Figure 5). The park is celebrated for its mile-long gorge, which is generally 150–200 feet deep but only 50–70 feet wide. The gorge was formed by Glen Creek, which runs for seven miles from west to east into Seneca Lake.

Watkins Glen's primary entrance is on the east, where Glen Creek interfaces with the Village (see Figure 6). The gorge starts immediately to the west of the main entrance and climbs in elevation by several hundred feet. Winding walkways, steps, and bridges take visitors through the rocky enclosure. The height of the sedimentary rock walls, known for their horizontal striations and dark gray color, frequently causes the only walkway through the glen, the Gorge Trail, to be shady and wet. In contrast to the Gorge Trail, two paved trails—the North Rim Trail and South Rim Trail—run along the gorge's rim, offering a more efficient and frequently less-crowded option to return to the main entrance. The park extends to the west of the upper entrance for nearly two and a half miles, though most of that space has limited improvements or is dedicated to organized group campgrounds and thus is not visited by most users.

### 5.2. Resort Era: 1863–1899

What is now Watkins Glen State Park was opened as a private destination on 4 July 1863. The idea for the destination came from its first manager, a newspaper editor from Vermont named Morvalden Ells. He named the site Freer's Glen after the property owner George Freer, though he changed the name to Watkins Glen in 1869 in honor of an early white settler, Dr. Samuel Watkins [27]. Despite opening during the middle of the American Civil War, the Watkins Glen had over 10,000 visitors its first year [28].

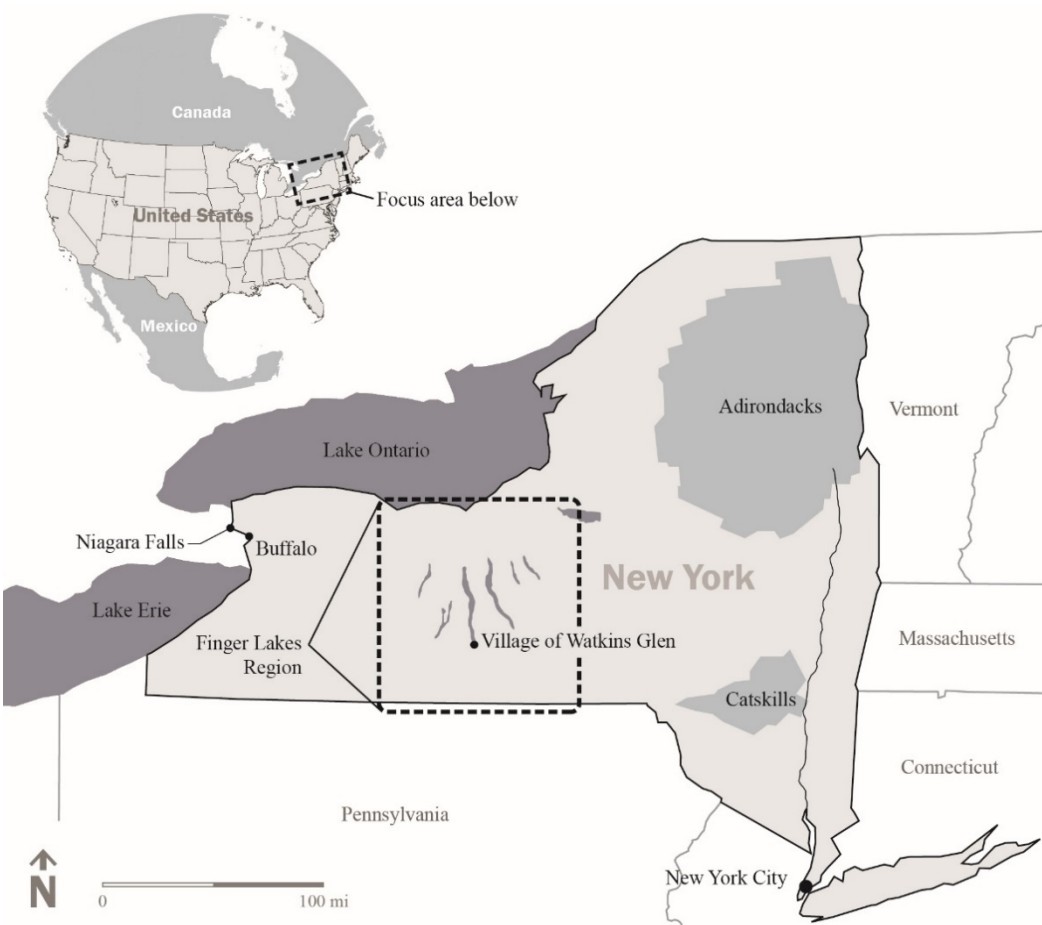

**Figure 5.** Map of New York State with North American context, by author.

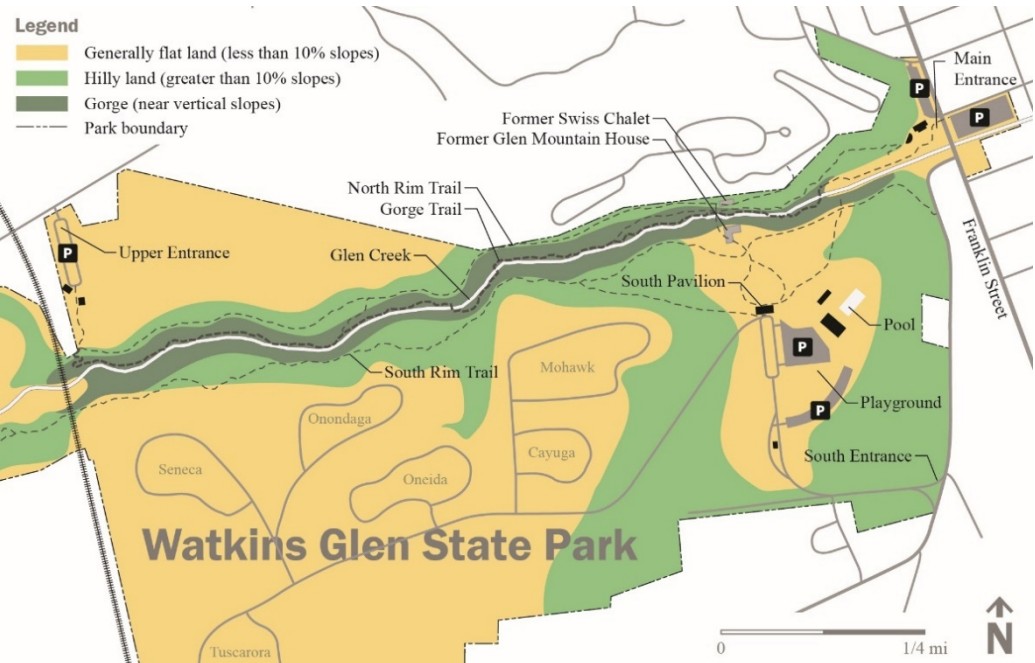

**Figure 6.** Map of the east side of Watkins Glen State Park in 2020, by author. Topographic information based on United States Geological Survey 2019 Quadrangle Maps.

Ells made three significant improvements that had a lasting impact on the park. First, Ells made the glen navigable by chipping narrow paths into the stone walls and, in hazardous spots, lining the path with railings made of branches. He also installed crude wooden stairs and bridges where pathways were otherwise impossible (see Figure 7). These improvements were needed; several newspaper articles noted that even with the additions, visitors should don weatherproof gear and sturdy shoes to negotiate the wet and narrow pathways [29]. Second, Ells built Evergreen, a Swiss-chalet-style concession building, atop a cliff, midway through the Gorge Trail [30]. Its Swiss façade likely aimed to connect the site with the Swiss Alps, which at the time were one of the most popular sublime landscape destinations in the world [31]. By 1867, Evergreen was replaced by an expanded concession and dining facility called the Glen Mountain House, which also had a Swiss façade [32]. Third, Ells named various spaces, objects, and views within the glen, with most names referencing biblical or mythological themes. The mile-long gorge walk was subdivided into eight different named glens, starting with Glen Alpha and ending with Glen Omega. Other objects in the glen are called Minnehaha Falls, Fairy Cascade, Neptune's Pool, The Labyrinth, The Grotto, Baptismal Font, Poet's Dream, Pluto Falls, and Pool of the Nymphs [28]. Naming places and scenes and dispensing those naming conventions in guidebooks were standard practices and suggested that Ells aimed to attract middle-class tourists [33].

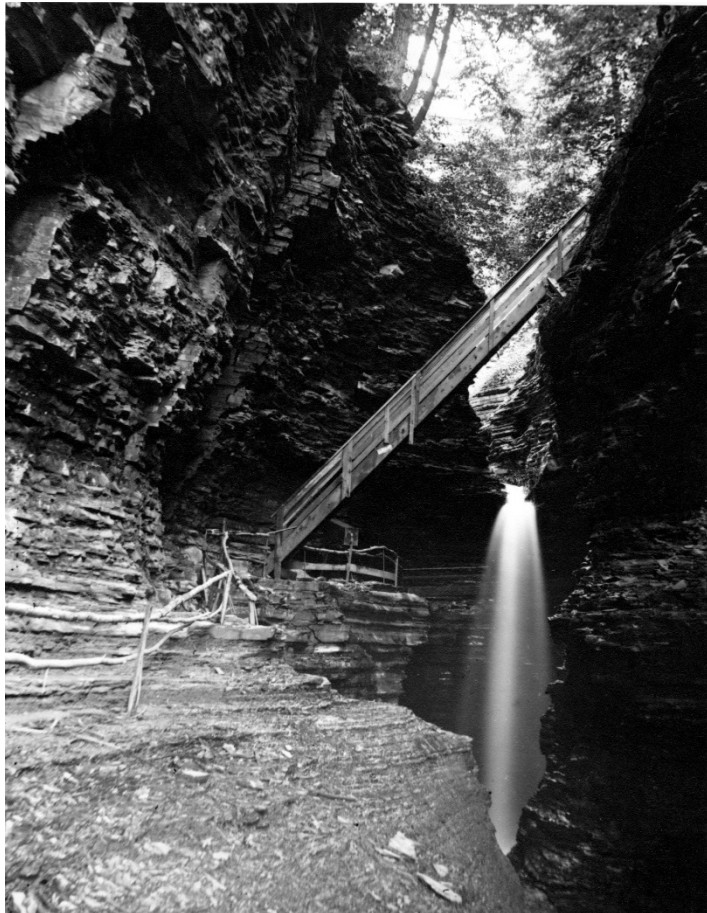

**Figure 7.** Cavern Cascade, ca. 1890. Photo used with permission: NY State Office of Parks, Recreation & Historic Preservation; parks.ny.gov (accessed on 12 June 2019).

While the Village of Watkins Glen was home to numerous hotels, the first and only hotel built in the park was constructed in 1873. The hotel took the name Glen Mountain House from the dining facility, subsequently called the Swiss Chalet. Connecting the two structures was a new iron suspension bridge, daringly crossing the chasm 100 feet above

the glen floor. The hotel was expanded in 1882 to include a billiard parlor, bowling alley, and music hall (see Figure 8) [34].

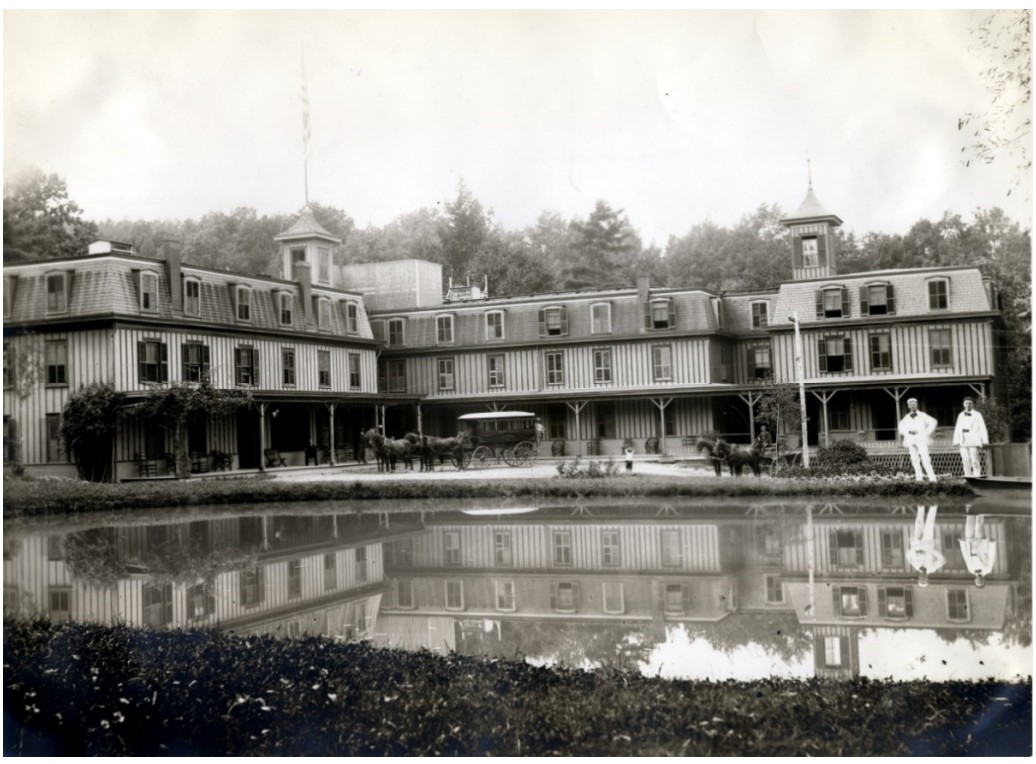

**Figure 8.** The Glen Mountain House, ca. 1880. Photo used with permission: NY State Office of Parks, Recreation & Historic Preservation; parks.ny.gov (accessed on 12 June 2019).

The Glen Mountain House's popularity began to wane towards the end of the nineteenth century due to several concurrent socioeconomic and recreation preference shifts. Working-class tourists, who started to have reliable free time on Saturdays, were looking for inexpensive destinations for day or weekend trips. In response, railroad companies offered single-day, round-trip tour packages to major destinations like Watkins Glen [35]. The new class of tourists wanted to experience scenic landscapes in different ways. Instead of being enraptured by the mythological associations promoted in guidebooks, working-class tourists wanted to inexpensively experience nature through forms of outdoor recreation such as hiking and camping [36]. Simultaneously, national centennial celebrations in 1876 made America's scenic landscapes icons of nostalgia and patriotism. This caused working- and middle-class tourists to question why so many of the best destinations were privately owned, over-commercialized, or made exclusive to only wealthy tourists. Municipal parks, such as Central Park in New York City, and a steady drumbeat of new national parks, such as Yosemite and Yellowstone, were making solid arguments that scenic landscapes were icons of America and thus should be accessible to all [37–39].

*5.3. Olmstedian Era: 1899–1923*

The first calls for public ownership at Watkins Glen were made in 1899 [40]. The inspiration for the requests came from Niagara Falls, which was transformed from an over-developed spectacle to a genteel public park fourteen years earlier [40–42]. The process to transfer Watkins Glen into public control took several years and included a reported assassination attempt, but it was ultimately successful. In 1906, Watkins Glen Reservation was opened to the public, free of charge. Although the park was controlled by the newly created Commissioners of Watkins Glen Reservation, the property was managed by the American Scenic and Historic Preservation Society (ASHPS) [43].

Between 1906 and 1912, the ASHPS made extensive aesthetic and programmatic changes to the park to demonstrate its ownership change. The Glen Mountain House and Swiss Chalet were both razed, as hotel and dining facilities were no longer appropriate or necessary in a public park. Inside the glen, the 6- to 24-inch-wide paths were replaced by 3- to 4-foot-wide walkways, and all wooden stairs and bridges were replaced with cast-in-place concrete [44].

The ASHPS focused most of their attention on the park's main entrance, where most new landscape elements mimicked those found at parks designed by Frederick Law Olmsted. This aesthetic dedication was with good reason: Olmsted was easily the most recognized landscape architect in the United States, known for his designs at Central Park and Prospect Park in New York City and designing the new public park at Niagara Falls [42,45]. Olmsted's public park designs often employed the picturesque aesthetic, where highly detailed built elements such as buildings, shelters, and bridges were placed in seemingly natural landscapes to evoke a sense of romantic wonder [46]. While neither Olmsted nor his firm designed the public amenities at Watkins Glen, the Olmstedian aesthetic is evident. The main entrance featured a stone gateway arch over the entrance drive, a symmetrical stone staircase into the glen, a bandstand for small concerts, and a Victorian-style water feature (see Figure 9). The most significant structure was the Entrance Pavilion, an Arts-and-Crafts style structure which housed a welcome center and restrooms. The amenities with the most direct Olmsted-lineage were the new iron railings—over 35,000 linear feet of them—placed along all pathways, stairs, and bridges in the glen (see Figure 10a,b) [44]. The "iron railings on inward-curving standards" were direct copies of the Olmsted-designed railings found at Niagara Falls [47].

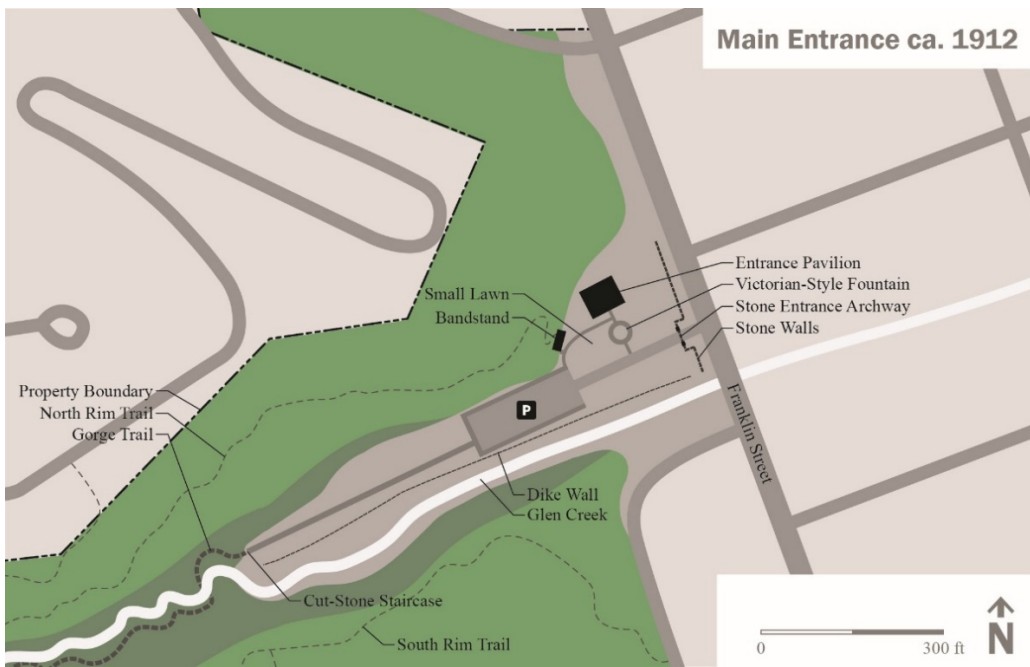

**Figure 9.** Plan of Main Entrance, ca. 1912, by author.

### 5.4. Rustic Era: 1923–1945

A new, rustic aesthetic in park architecture appeared in the early twentieth century. This "natural" aesthetic was born out of the Arts and Crafts era, which valued hand-crafted finishes, but was amplified and modernized due to the immense scale of national park hotels. Old Faithful Inn at Yellowstone National Park and El Tovar at Grand Canyon National Park, constructed in 1903 and 1905, respectively, used large-scale natural materials such as boulders and logs to make the architecture blend into the landscape and, at times, even make the architecture appear to grow out of the landscape [37]. The railroad

companies that built those hotels used the unique architecture in their advertisements, which began to set a national standard for park architecture.

At Watkins Glen, the rustic aesthetic appeared in the park in 1927 in the South Pavilion, a picnic and restroom facility built on the south side of the glen (see Figure 11) [48]. The structure was needed; since the Glen Mountain House and Swiss Chalet were razed in 1908, there were only two restroom facilities in the park, one at the park's main entrance and one at what would become the Upper Entrance [49]. That left the campsites, all located in the middle of the park, without restroom facilities.

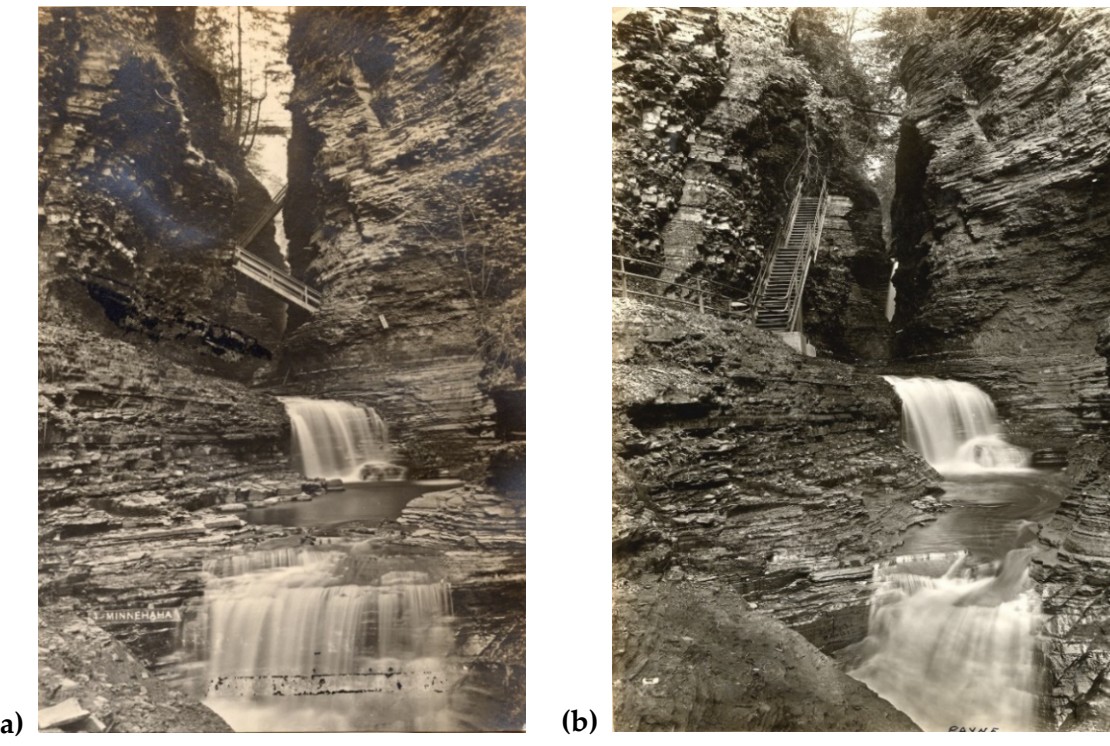

**(a)** **(b)**

**Figure 10.** Minnehaha Falls ca. 1890 (**a**) and ca. 1920 (**b**) Photos used with permission: NY State Office of Parks, Recreation & Historic Preservation; parks.ny.gov (accessed on 12 June 2019).

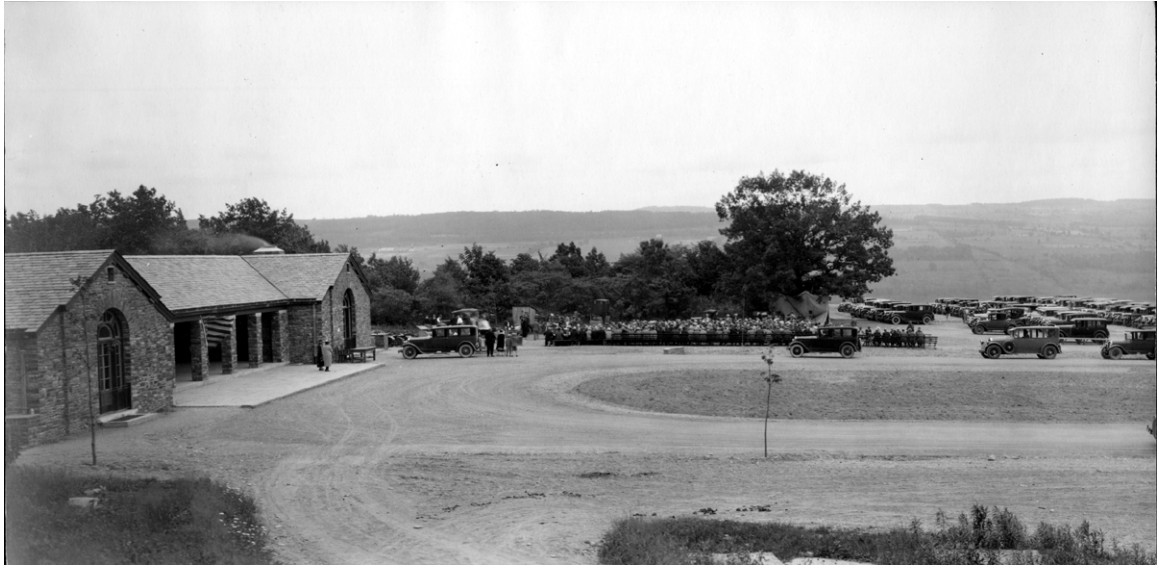

**Figure 11.** South Pavilion and South Entrance Parking Lot, ca. 1927. Photo used with permission: NY State Office of Parks, Recreation & Historic Preservation; parks.ny.gov (accessed on 12 June 2019).

Adjacent to the South Pavilion, an additional 400 acres of land was acquired to allow for more parking [50]. Before the addition of the South Pavilion, the parking lot at the main entrance accommodated around 100 cars but needed space for 1000 vehicles, which were otherwise parking on the surrounding streets [51]. Since the main entrance parking lot had limited expansion potential due to its position between a cliff and a creek, the decision was made to build a new, 1000 stall parking lot adjacent to the South Pavilion. A small amount of additional parking was added at a new Upper Entrance, found at the glen's west end [52].

The Great Depression significantly impacted both the number of tourists and the amount of state funding available for park improvements. In 1933, all state parks in the Finger Lakes Region began charging a 25-cent fee to park vehicles on the property [53]. To make capital improvements, the New York State Park System applied to the newly created Civilian Conservation Corps (CCC) state park program for assistance. The CCC was created in 1933 to put unemployed young men to work and perform conservation duties to improve soil and forest qualities across the country [54]. In 1935, CCC camp SP-44, under the National Park Service's supervision, was opened on the west end of Watkins Glen [55].

The CCC men were tasked with continuing the work that park staff started in 1930, reversing the Olmstedian aesthetic inside the glen by cladding the concrete bridges with natural stone. Their work had only just begun when the park landscape changed dramatically. In July of 1935, heavy rains caused a log jam to break west of the park, sending a tumult of water through the glen. When it was over, almost all the railings and bridges in the gorge were swept away. The floodwaters removed much of the parking lot, stone gateway, and walls at the main entrance (see Figures 12 and 13) [34]. With a relatively clean slate, the CCC men began to rebuild the bridges, stairs, and pathways out of stone instead of concrete. Instead of Olmsted's railings, low stone walls were added where necessary to keep visitors safely on the paths. At the main entrance, the stone gateway was removed and replaced with rustic stone columns, like those found at other state and national parks (see Figure 14) [56]. Additionally, a new restroom build was built just to the west of the Entrance Pavilion and parking lot was expanded to accommodate more vehicles.

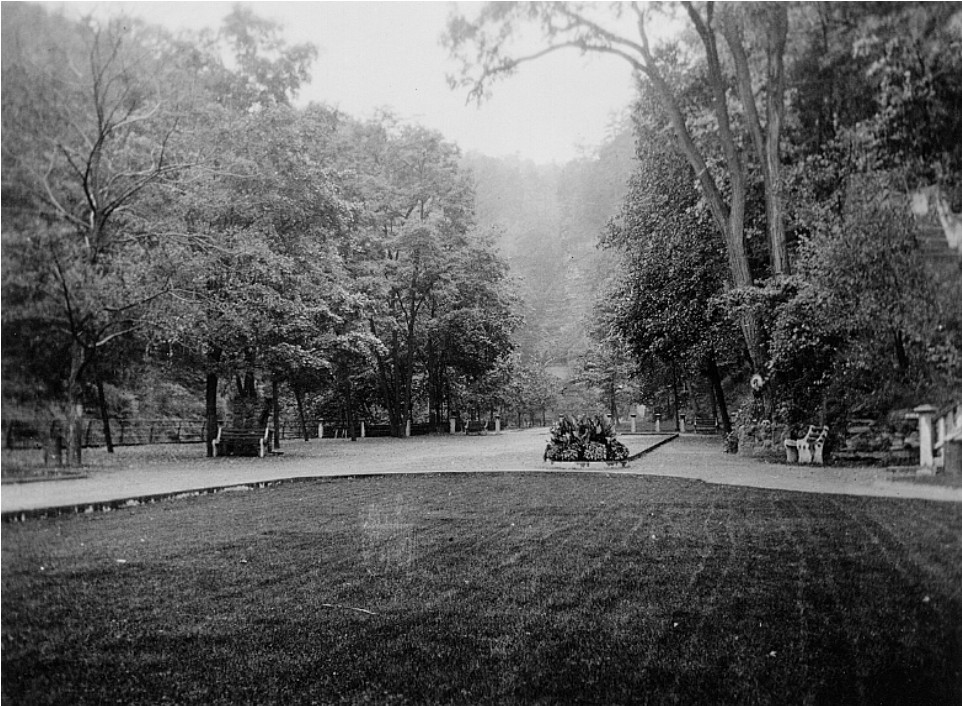

**Figure 12.** Entrance Lawn, ca. 1920; before the flood of 1935. Photo used with permission: NY State Office of Parks, Recreation & Historic Preservation; parks.ny.gov (accessed on 12 June 2019).

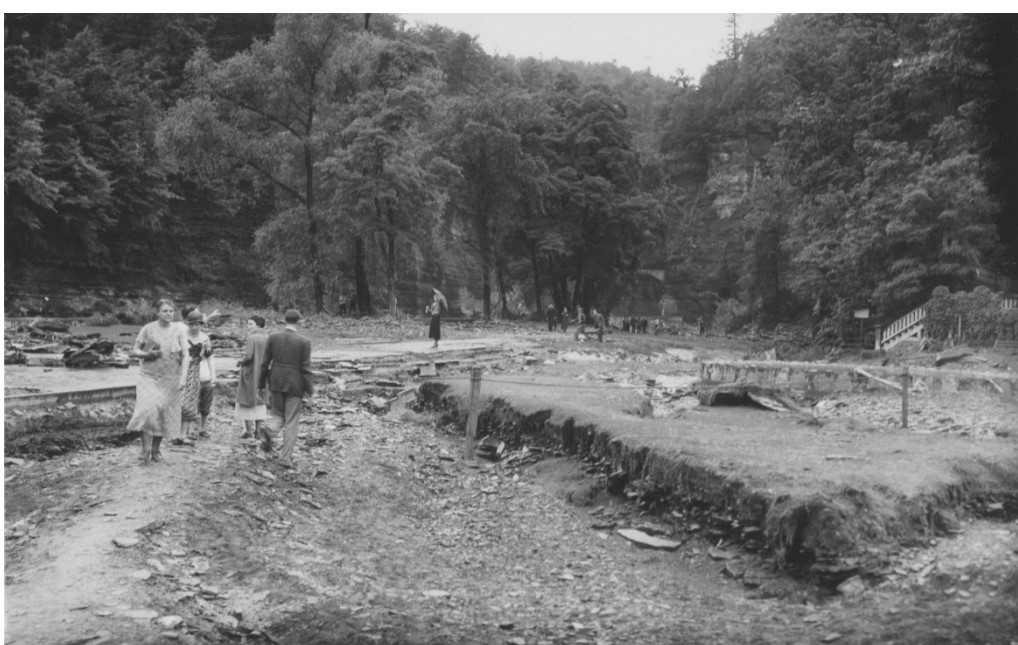

**Figure 13.** Entrance Lawn after the flood, July 1935. Photo used with permission: NY State Office of Parks, Recreation & Historic Preservation; parks.ny.gov (accessed on 12 June 2019).

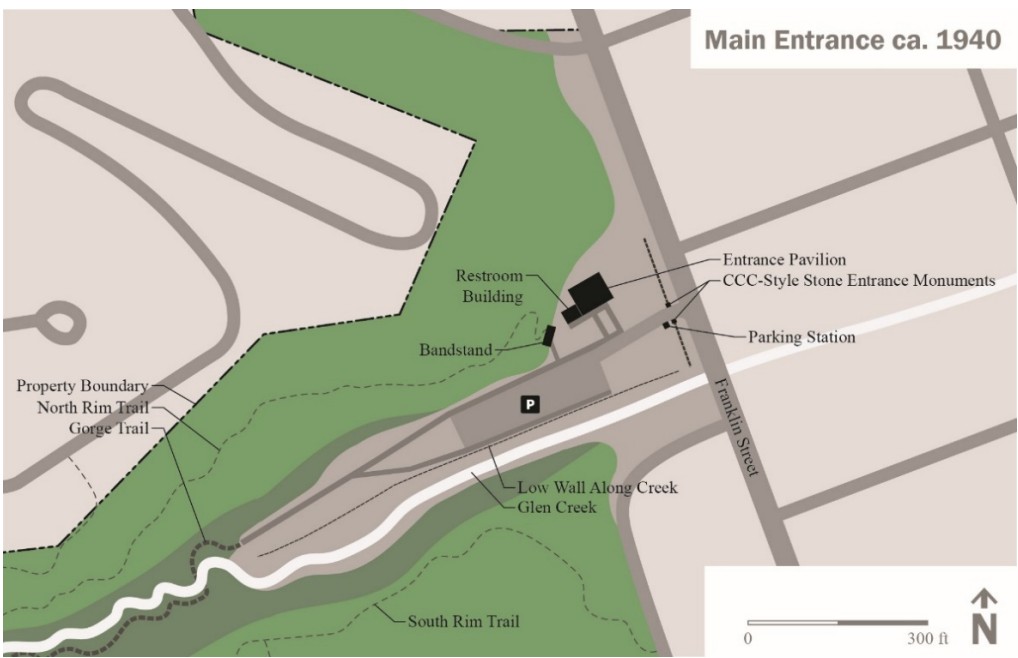

**Figure 14.** Plan of Main Entrance, ca. 1940, by author.

### 5.5. Entertaining Era: 1945–1970

World War II and the post-war economic recovery slowed the tourism industry in the United States, and thus, few changes were made at Watkins Glen during the 1940s. That changed in the early 1950s when families traveling in station wagons began to dominate the tourism market [31]. In response, the managers at Watkins Glen started diversifying amenities to keep the entire family entertained. In the late 1950s, a large playground was built near the South Pavilion [57]. The need for the playground was due in part to the increasing number of tourists who were coming to the area for automobile races. In 1948, a racing enthusiast named Cameron Argetsinger started a 6.6-mile automobile race that circumnavigated Watkins Glen State Park. While the race eventually moved to a closed

course southeast of the park, the Watkins Glen Grand Prix drew tens of thousands of racing fans to the area, boosting the park's notoriety and visitation. The track's popularity reached a pinnacle in 1973 when a summer concert held at the racetrack drew 650,000 spectators [58]. Understandably, visitation to Watkins Glen State Park soared as tourists looked for inexpensive options to extend their stay in the Finger Lakes Region.

To attract and keep those visitors, a multitude of entertainment options were added in the park. In 1963, an Olympic-sized swimming pool was built north of the playground. The pool was first proposed in 1960 by community members who aimed to attract more tourists through expanded recreation options [59]. By this time, other state parks in the Finger Lakes Region were well known for their novel swimming options. Robert H. Treman State Park had a natural swimming pool at the base of a waterfall, and Taughannock Falls State Park had wide rocky creeks which were frequently used for swimming and creek walking. Watkins Glen had neither the natural facilities nor the space to accommodate such options, so the commissioners at Watkins Glen opted to build a swimming pool and make it notable due to its Olympic size. The pool opened to great fanfare on 22 June 1963—nearly 100 years to the day of the glen's original opening [60].

The new swimming pool displaced most of the park's campsites, so new camping facilities were designed. The new campgrounds, which opened in 1965, were organized around two looped drives, with dedicated parking stalls at each campsite for vehicles and trailers. The camping facilities were so popular that four additional loop drives with campsites were added in 1967 [61]. All looped drives were named after Native American tribes who lived in the region (see Figure 6). To entertain the hundreds of nightly campers, the park offered daily concerts and movie nights in the peak summer months of the late 1960s and 1970s. Other cultural events such as car shows, social club events, and beauty pageants were held in the park to keep visitors entertained [62].

*5.6. Interpreting Era: 1970–2018*

Environmental awareness and the recognition of environmental education in the late 1960s fostered a desire among park professionals to raise the public's consciousness about nature [63]. This educational approach to environmentalism was seen at Watkins Glen in the mid-1970s when the first full-time interpreter was hired to give presentations and guided hikes. In 1979, the Entrance Pavilion was expanded to accommodate new restrooms. The interior was then redesigned as a dedicated interpretive center. Additionally, dozens of interpretive signs were added throughout the park, describing the glen's geology, flora, fauna, and history [64].

The park's popularity changed dramatically in 1980 when the nearby racetrack lost hosting privileges of the US Grand Prix. The track closed entirely for two years, starting in 1982, which significantly impacted tourism in the Finger Lakes Region. To attract new visitors, Watkins Glen added a new interpretive amenity in 1983: Timespell, a light-and-laser show which traced the geologic history of the gorge by projecting images and lasers on the walls of Glen Alpha. Like the swimming pool, Timespell was explicitly created to extend the visitor's stay [65].

To accommodate the new show, the main entrance to the park was redesigned once again (see Figure 15). The CCC-era stone walls and monuments were replaced with an open lawn, ultimately blurring the boundary between park and community. The Victorian fountain was removed and replaced with an expansive concrete plaza and a new Timespell-specific ticket booth. The bandstand was transformed into a concession stand, offering wine and cheese for its evening visitors [66]. An additional parking lot was installed north of the main entrance, adding parking capacity for approximately seventy additional vehicles, including recreational vehicles (RVs). While Timespell was widely popular for a few years, its attractiveness waned as its technology became dated. Interpretive efforts took a further hit in 1995 when state-wide budget cuts significantly curtailed interpretive efforts in all parks [67]. While guided tours were still offered, most other interpretive programs stopped. Timespell offered its last show in 2001 [65].

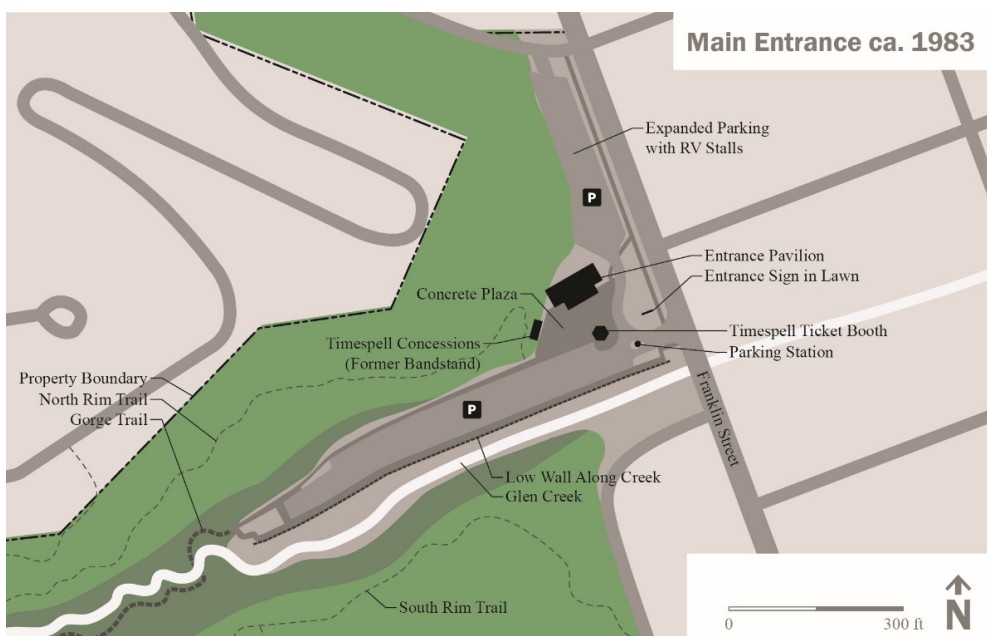

**Figure 15.** Plan of Main Entrance, ca. 1983, by author.

*5.7. Urban Era: 2018-Present*

　　The last significant change to happen at Watkins Glen involved the complete redesign of the main entrance in 2018, after thirty-five years without substantial improvements to the park (see Figure 16). The most significant functional change was the relocation of the parking lot at the main entrance, which was moved to the east side of Franklin Street on newly acquired property (see Figure 6). Additionally, all Timespell infrastructure, which had been sitting unused for sixteen years, was removed. With the parking and ticket booth gone, most of the space at the main entrance was dedicated to pedestrians, including open lawns, a small amphitheater, a wide promenade, and paved public plazas. The bandstand/concession structure was removed and replaced with a new Visitor's Center, jointly owned by the park and the Village of Watkins Glen. The Entrance Pavilion remained as a gift shop and interpretive center. Festooning the entrance once again were low stone walls, refined versions of the CCC-era walls which once lined the entrance [61].

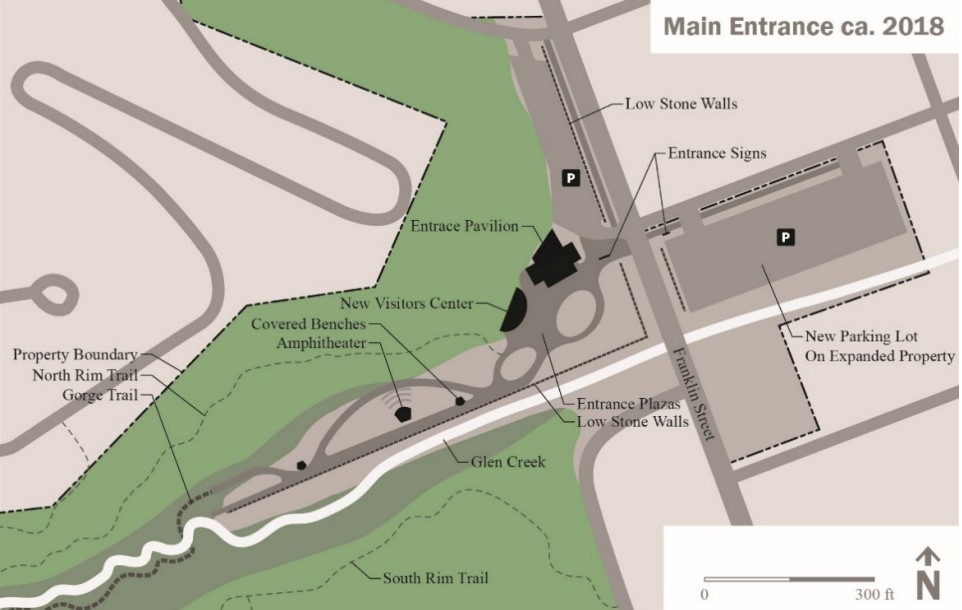

**Figure 16.** Plan of Main Entrance, ca. 2018, by author.

The amenities and aesthetics of the 2018 redesign contain a mix of historical and contemporary program elements and styles. Many structures and spaces, including the new Visitors Center and plaza areas, have adopted materials and forms found in contemporary urban parks: glass curtain walls, stone cladding, and clean curvilinear retaining walls. Other amenities recall the past: the promenade and open lawns recall Olmstedian parks, while the oversized lumber and stone façade of the amphitheater recall CCC structures from the 1930s. Some scholars have argued that this mixed aesthetic is becoming common in urban parks, which attempt to anticipate the needs of as many people as possible [39]. The redesign at Watkins Glen is too recent to understand its inspiration and evaluate its success. However, its amenities and aesthetics suggest that park managers and designers opted to keep pace with current urban trends in park design.

### 5.8. Summary of Changes at Watkins Glen

The most common reason the designed landscape at Watkins Glen changed was to ensure its amenities and aesthetics could be associated with other scenic landscape destinations (see Figure 17). In some instances, that association was performed through aesthetic decisions such as the Swiss façade of the Swiss Chalet, Olmsted-inspired railings, and the South Pavilion's rustic exterior. In other cases, amenities were added or removed to maintain associations with other destinations. Evidence for this can be seen in the addition of the billiard and music rooms to the Glen Mountain House and, later, the playground and swimming pool.

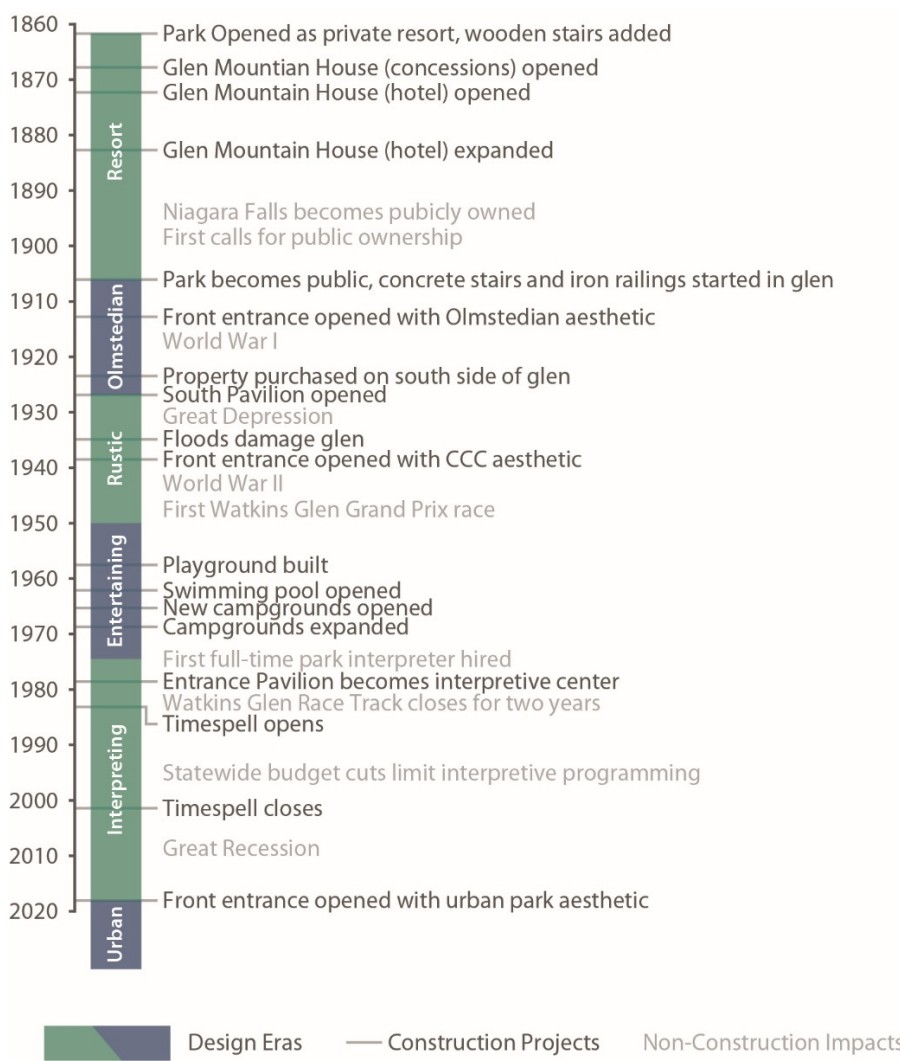

**Figure 17.** Summary of Design Changes at Watkins Glen, by author.

## 6. Results

### 6.1. Watkins Glen and Butler's TALC Model

Watkins Glen experienced many of the TALC model stages, but, like many publicly owned natural destinations, the park has maintained a stable maturity and popularity [16]. The park's "exploration" stage occurred before 1863, when the destination was known as a local curiosity. The "involvement" and "development" stages then co-occurred between 1863 and 1880. As the destination's scale and community scale were small, most residents were invested in the resort concept from the start. While most of the early managers were external investors, the managers had to lease the property from local landowners, so local involvement was inherent.

Evidence of "consolidation" and "stagnation" stages are difficult to find at Watkins Glen, and the "decline" stage is absent. The best evidence of stagnation would be the various attempts to draw more users through expanded park activities, such as the campground, swimming pool, and Timespell. These were clear examples of adding new amenities to counteract a slowdown or stagnation in attendance. However, these additions were implemented over eighty years, so it is difficult to attribute their addition to a single stage.

One reason for this lack of consolidation and stagnation is likely due to the value shifts made when the park transitioned from private to public ownership. As a privately owned tourist destination, the glen's primary purpose was profitability for managers, landowners, and surrounding businesses. As a public park, Watkins Glen became a social and cultural asset that reflected regional and national identity aspirations. While still tied to the capitalist marketplace through surrounding businesses, economic profit was not the park's sole focus. Therefore, the traditional TALC model was bound to be interrupted or altered.

Importantly, carrying capacity was rarely ever exceeded, at least in the way the TALC model suggests. The closest example of exceeded capacity involved the glen's paths, stairs, and railings which were upgraded in the 1910s. While the paths were widened to accommodate more visitors, there is no evidence that complaints spurred the change. Instead, the impetus was likely liability concerns over the narrow, uneven walkways and unstable railings. The other changes spurred by carrying capacity were the addition and expansion of parking lots in the 1920s and the expansion of camping facilities in the 1960s. These changes were caused by demand, not overcrowding, so while they involve carrying capacity, they struggle to conform to the TALC's definitions.

### 6.2. Watkins Glen and Agarwal's Model

It is easier to consider Watkins Glen's prolonged maturity as a series of Agarwal's "reorientation" stages [20]. After reaching a state of maturity, destination managers at Watkins Glen routinely used the addition or subtraction of amenities and aesthetic styles to reorient the park to meet visitors' desires. The best example is the reorientation that occurred when the park transitioned to public ownership. The resort amenities were replaced with campgrounds and picnic shelters to reposition the destination as an egalitarian public park rather than an exclusive retreat. However, using the TALC as a foundation, Agarwal inherently employs carrying capacity as the primary instigator of change, which, as noted above, was not a significant driver of change at Watkins Glen. Additionally, the "physical plant" at Watkins Glen—the glen and its scenery—has not been substantively changed since 1935. This lack of change gives credence to Agarwal's 2006 discussion of rejuvenation strategies, particularly "repositioning," when the attraction source remains the same but external amenities are revised to attract different clientele [68].

### 6.3. Watkins Glen and Haraldsson and Ólafsdóttir's Model

Watkins Glen has similarities with Haraldsson and Ólafsdóttir's model, though, like Agarwal's model, it is not easy to find moments where visitors overwhelmed the destination's carrying capacity. There is evidence of a change in the visitor's preference for nature purity, though Watkins Glen oddly rearranges Haraldsson and Ólafsdóttir's development

suggestions (see Figure 3). Watkins Glen was created to be a middle-class resort, complete with music halls and billiard rooms. However, it became more natural because purists felt scenic landscape destinations should be publicly owned. From that point, the park's development trend has generally followed Haraldsson and Ólafsdóttir's development model, as more infrastructure and amenities were added to accommodate and attract as many tourists as possible [24].

Watkins Glen also challenges Haraldsson and Ólafsdóttir's model by highlighting the importance of developmental revision. Though aesthetically appropriate for the time, the concrete stairs and iron railings added to the glen between 1906 and 1912 were quickly viewed as a design mistake. Carl Crandall, the chief engineer of the Finger Lakes State Park Commission between 1924 and 1961, said of the stairs and railings, "The flood swept away most of the massive and alien architecture installed by the state in early days before the minds of the authorities had acquired the taste and discrimination which demand that construction work be fitted to the natural appearance of the surroundings" [69]. The removal of resort facilities and revision of stair infrastructure illustrates that Haraldsson and Ólafsdóttir's model is not inevitable. Indeed, the ability to remove unwanted infrastructure may suggest a prolonged maturity stage in the nature purity model, where destinations revise their amenities and infrastructure to continually stay in check with visitors' preferred urban/nature balance.

In summary, Watkins Glen does not fit any of the prevailing tourism life cycle models, primarily because it transitioned from private to public ownership and rarely exceeded its carrying capacity. Instead, the park adapted to changes in tourists and tourist expectations by continually revising its amenities. While the "physical plant" for the destination—the glen—has remained constant, park managers used designed amenities to adapt the destination's concept to meet its users' demands.

### 6.4. Watkins Glen and the Concept Renewal Cycle

At Watkins Glen, the "concept creation" phase primarily took place between 1862, the year before the park opened, and 1873, after the Glen Mountain House was established (see Figure 18). The created concept was focused on relaxation and socialization in nature and therefore targeted an upper-middle-class audience.

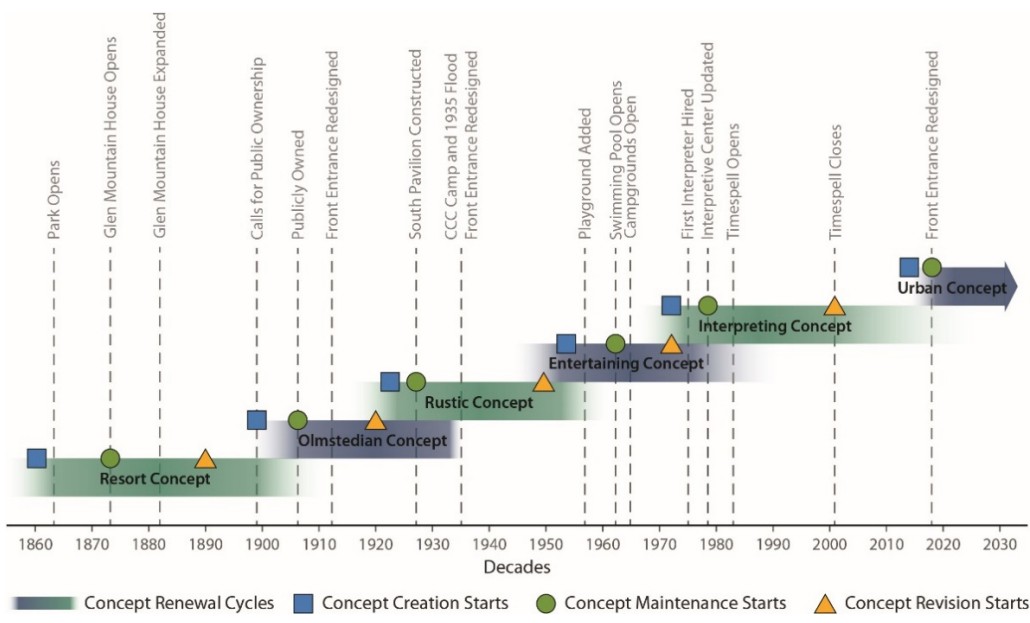

**Figure 18.** Concept Renewal Cycles at Watkins Glen State Park. Source: Author.

The first "concept maintenance" phase at Watkins Glen occurred between 1873 and 1895. During this era, the middle-class resort concept was reinforced and repositioned

to ensure its amenities were meeting demands. Glen Mountain House was expanded to include billiard rooms and bowling alleys, images of the glen were created, and advertisements continued to extoll the site's unique amenities. These efforts underpinned the concept of a middle-class relaxation destination, but they did not push Watkins Glen to exceed its carrying capacity nor cause it to decline; they merely reinforced the established concept.

The first "concept maintenance" phase at Watkins Glen came to an end due to the rise in working-class tourists, who could inexpensively travel to the glen and back without needing overnight accommodations. The gradual push for public ownership, spurred largely by the opening of the public park at Niagara Falls in 1885, also challenged the middle-class resort concept.

At Watkins Glen, the second CRC started in 1899 when the first calls for public ownership were made. When the park was opened to the public as a state reservation in 1906, its new park managers quickly developed and installed a destination concept that made the park feel and appear public. They opted to utilize the Olmstedian aesthetic found at Niagara Falls and Central Park, so the park was visually and functionally similar to other popular public parks. The third cycle started at Watkins Glen in 1923 when the first plans were developed for the stone-clad South Pavilion—only thirteen years after the Olmstedian iron railings were installed. The concept was solidified after the flood of 1935 when most of the park infrastructure adopted the CCC aesthetic.

A fourth CRC began in the late 1950s when the park concept was pivoted toward entertaining families on vacations, especially racing fans. In response, the playground, swimming pool, and campgrounds were added, and movie nights and concerts kept camping guests entertained throughout the summer months. The fifth CRC began in the mid-1970s when the concept of Watkins Glen shifted to accommodate the desire for natural and cultural interpretation. While expectations for entertainment remained, the park was transformed to literally place interpretation and education at the forefront, as evidenced by the addition of the Timespell infrastructure at the main entrance. The sixth and latest CRC began in 2017 when funds were again allocated to redesign the main entrance. In this change, the goals were to remove automobiles, dedicate space to pedestrians, and offer as many park resources as possible, goals similar to many other urban parks.

## 7. Discussion: Concept Renewal Cycles at Watkins Glen

When viewed through its numerous conceptual cycles, the case study of Watkins Glen illustrates that the designed landscape, including its arrangement of amenities, structures, materials, and ornamentation, is an essential part of understanding the evolution of tourist destinations. The landscape is the palimpsestic display of the destination's concept. It is the collection of objects and spaces that subconsciously and physically constitute the destination and its attractions. The static landscape image forms the "markers" (e.g., postcards, guidebook images, images on keychains) that tourists use to determine value, appropriateness, and relevance [26]. The landscape is also the dynamic medium that alters tourist behaviors, reactions, and consumptive processes. The dynamic landscape is in a constant state of change, as tourists continually evaluate the concept, other tourists, and their role as tourists in that space [11]. The output from the dynamic landscape, the subconscious conceptualization and digestion of the activities, is encoded as part of the static concept, further differentiating and assimilating the destination in the tourism marketplace.

The Olmstedian and urban concept cycles at Watkins Glen illustrate that a destination's concept can rapidly change, provided most designed elements display that cohesive conceptual theme. In those cycles, the main entrance was wholly redesigned using recognizable amenities and aesthetics, at least for the clientele at the time. The accessibility of known landscape tropes allowed tourists to decipher the landscape quickly, evaluate its concept, and assess their social relationship with the destination. However, the financial implications and risk to make such comprehensive changes were significant.

Large-scale landscape change requires considerable financial investment, further relying on understanding the tourism marketplace by managers and their hired professionals. If the managers' prognostication is incorrect, the significant investment may not result in a concept relevant to their preferred clientele.

The location of that investment may suggest that tourists utilize the visual and performative concept shown through the landscape at the main entrance to determine whether to visit the destination, then further refine their evaluation as the totality of the destination is experienced. This idea of design concept digestion and evaluation deserves more inquiry, as it impacts both tourists and managers.

In contrast to the rapid concept changes, the entertainment and interpretation concept cycles at Watkins Glen suggest that concept cycles can also change very slowly, overlap, or seamlessly blend. Between 1927 and 1935, the park offered both Olmstedian and rustic styles, albeit in different locations of the park. The main entrance would have likely retained its late-Victorian style for considerably longer if not for the flood of 1935 that demanded new infrastructure. The concurrence of styles allowed the park to transition between concepts slowly. This was seen in the 1970s as well, when interpretive programming was offered alongside the entertaining movie nights. Instead of starkly ending one concept, managers at Watkins Glen chose to slowly wind down the more overt entertainment options in favor of more educational programming, therefore aligning the park concept with other education-focused parks of its time. This overlap or gentle transition in cycles highlights the continual performative role that the landscape plays. The managers' daily decisions changed how visitors engaged with the destination, and therefore incrementally changed the park concept.

Concepts can also stagnate, as evidenced by the prolonged interpretation concept cycle that languished into the 2000s and 2010s. Unlike the TALC's "stagnation" stage, which focuses on the stagnation of visitation and economic growth, conceptual stagnation happens when the destination fails to evolve with its tourists [13]. While Watkins Glen did not decline due to its stagnant concept, other private destinations may not have fared so well. A stagnant or out-of-date concept may signify a lack of investment in the destination, which illustrates the connection between the concept and financial capital. Without money, the landscape cannot change, and thus the concept cannot change. It must be noted that this study did not focus on financial investment and its impact on visitation, so the connection between concept stagnation and capital investment are correlations only. Regardless, this is an area that deserves further research, as the potential tie between concept and financial investment has considerable implications on the evolution of tourist destinations.

Another topic that deserves additional research involves the location of developments and the impacts of those developments on ecosystem quality. As Figure 6 shows, most new infrastructure and amenities at Watkins Glen were added where the natural topography was generally flat and therefore conducive to construction. Conversely, developments in the steeper areas of the property and gorge have been limited to trail networks which allow users to navigate those areas of the site safely. While the Glen Trail's aesthetics have been controversial at times, the gorge has not effectively changed since the late-1930s. The property's peripheral and flat areas have been the locations for change, which has undoubtedly impacted ecosystem quality. This brings up several rhetorical questions that challenge the CRC. How much of the site's longevity is due to the general preservation of the gorge? If the glen would have been radically and irreparably altered, say with a restaurant built into the gorge walls, could new concepts rectify such an intrusion? Might this suggest a "point of no return" in terms of intrusive landscape change? Is there a tolerance for landscape change and ecosystem degradation in external areas of natural tourist destinations, provided the primary tourism product is preserved? Again, these areas deserve additional research to understand their impact on the tourism destination and the CRC as a longevity model.

## 8. Conclusions

The Concept Renewal Cycle provides a theoretical framework upon which design changes are organized to understand destination evolution. Its cycle of "concept creation," "concept maintenance," and "concept revision" phases rely on relevance as the criteria for change. Watkins Glen State Park was used as a case study because it did not fit well into many other prominent life cycle models yet has remained a popular scenic landscape destination for nearly 160 years. The CRC helps explain its longevity by illustrating that a destination's concept and its concept management are essential for long-term success.

The CRC's purpose is not to replace other life cycle models, as it does not include the essential economic or visitation variables that illustrate growth and decline. Instead, the CRC should be used to expand how destinations are conceptualized and understood. At its core, the CRC is a non-linear analytical process that illustrates how and why destinations change. That information can then determine the applicability of other life cycle models. Hopefully, its continued use will progress efforts made among design and planning researchers to elevate the importance of design in tourism scholarship. In turn, it will ideally foster more communication and collaboration between designers, managers, economists, geographers, and other tourism specialists.

Notably, the CRC is a tool that managers can use to make future management decisions. Managers should use the CRC as a destination roadmap by understanding where on the cycle their destination sits. That knowledge can then clarify what options are available, including the ability to prolong or reconsider their concept and illuminate ways to make meaningful change. As the CRC illustrates the landscape's role as the static display that visually projects the destination's concept, managers should manipulate the landscape through the addition or subtraction of amenities, materials, and aesthetic styles to ensure it displays the intended concept. The CRC also illustrates that the landscape is the dynamic medium that affects change. Therefore, managers should utilize the landscape to choreograph tourist behaviors and consumption, ensuring the tourists' experience matches and continually reinforces the intended concept.

**Funding:** This research received no external funding.

**Acknowledgments:** The author would like to thank Hossein Entezari for aiding in graphic development and Heidi Hohmann and Michael Martin for their constructive comments. Additional thanks go to Josh Teeter and Fred Bonn from the New York State Office of Parks, Recreation, and Historic Preservation.

**Conflicts of Interest:** The author declares no conflict of interest.

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
