# Peer review of "Design as an Indicator of Tourist Destination Change: The Concept Renewal Cycle at Watkins Glen State Park"

_land, doi:10.3390/land10040367_

Round 1
Reviewer 1 Report
The work is interesting, it combines the subject of landscape design with the management of the tourist region. Despite the high value of the work, following improvements should be applied to, increase the quality of work:
- The objective should be identical and clearly formulated in the abstract and in the introduction; meanwhile, the reviewer only guesses at it based on the research questions formulated in section 1.2.
- The structure of the work, in my opinion, needs ordering. Basically, the following structure is assumed in scientific articles: introduction, materials and methods, literature review, results, discussion, summary and conclusions. Currently, the structure of the paper is very complex, the chapters are disproportionate in content (compare Chapter 1. and Chapter 2.). The results are presented in extensive chapters 3.-5., while the discussion is presented in chapter 6. and partly in the previous ones; I suggest rethinking the structure of the paper, e.g. combining and shortening or widening some chapters.
- In defining terms (currently chapter 1.3.) the term "tourist landscape" can be also defined.
- Chapter 2. needs to be expanded. It is necessary to describe the research procedure in detail; here also, in my opinion, there should be a justification of the choice of the case study with the characteristics of Watkins Glen State Park included in chapter 3.1. Maybe it is also worth to write about the tourist traffic in the park (number of visitors per year) and the types of landscape found in the park?
- The discussion needs ordering and enriched with references to literature. The author writes already in the abstract: "few life cycle models consider the designed landscape as a factor in the evolutionary process or as a signifier of change", but I did not notice references to specific items of literature presenting research results in other tourist regions.
- The characterisation of changes in the development of the area is mainly descriptive. The author enriches it with archival photographs. But maybe it should be considered to illustrate it with figures, showing the structure of the park in particular periods? As a summary of the characteristics it would be advisable to have a table presenting the introduced elements of management/changes, although they are presented to some extent in Fig. 14.
- Fig 4. needs improvement. The boundaries of the Finger Likes Region are perhaps incorrect; the map scale and map orientation (direction) are missing.
- Fig 5. shows only part of Watkins Glen State Park. Why isn't the whole area shown?
- Some unintelligible terms need clarification, such as "the landscape is the stage where the tourism performance takes place" [11-12] "The CRC illustrates the landscape's importance as the static and dynamic medium in which managers can affect change" [731-732].
- Linguistic correction of the entire text and confirmation of permission to reproduce photographs is advisable
Reviewer 2 Report
This paper proposest the model called Concept Renewal Cycle. The model has been succesfully applied to the the case study of Watkins 16 Glen State Park in New York state.
This is well structured and well written paper, which is easily readable also by non-experts in the field.
I have not any relevant comment to report.
It is my opinion that the paper can be accepted in the current form.
Reviewer 3 Report
The paper proposes a model called the Concept Renewal Cycle (CRC) to understand and document destination change of a tourism landscape. The text gives many definitions which makes clear the conceptual references of the research. Especially the first part is an interesting review on the literature on Tourism Life Cycle Literature. In the second part, the study uses archival research and the interpretation of graphic representations to produce a longitudinal review of changes of the landscape taken Watkins Glen State Park as an exemplar case study. The perspective consider some of the variables involved in the evolution/transformation of landscape focusing on historical and aestetic variables.
I have only a few remarks.
Lines 158-166 = the Methods are quite general and should be improved; it is not clear the set of documents analysed (newspapers? Consultant reports? Etc.), the authors/ competences/role of the reports used and the range of years considered in the research.
Differently, the Author can decide to change the title of this paragraph, and consequently also those of the ‘Results’: this article can have a style quite different from a research paper with ‘methods’ and ‘results’ .
Fig. 4 = the figure should be completed with a smal image of North America and a spot indicating the location of the analysed site.
Reviewer 4 Report
A very well-written, interesting and innovative paper. I enjoyed reading it!
Reviewer 5 Report
Design concept as an indicator of the tourism life cycle change seems a new analyses system in the conversation about change and adaptation at tourist destinations. Most life cycle models do not take care of the design concept as a factor of change. However, designed landscapes' attractiveness is a strong call in searching for tourism destinations.
The introduction is a clear overview of the regularly used life cycle models. The model site, the Watkins Glen State Park offered a 160years long design history from a private resort era to a public state reservation and up to entertaining driven developments. Still, as a result, the park's maturity escaped from turning into decay thanks to several redesign concepts.
The presentation of the research is clear, and well-debated though a bit wordy and lacks more detailed map analyses and figures. Figure 14 would fit better to point 3. where the concept renewal cycles are analysed based on publications and documents. I wonder if landscape character and ecosystem typologies could help in understanding the evolution and development of the various design concepts. Since carrying capacity and its threshold are key issues in the touristic areas' life cycles, mapping of the ecosystem and its sensitivity against development types and use intensity would be relevant to understand the absence of the "decline" stage.
Point 3. is the Result where Watkins Glen's designed landscape change was introduced. This is a descriptive part where the development is given in seven stages with two large-scaled maps only. Point 4. is the evaluation of the history and the development stages along with the existing tourist life cycle models. Then we have Point 5. which is the new method, the concept renewal cycle with the explanation of the terms and sections, and with some repetitions of the historic analyses. The irregular structure, the results in three main parts and the sort os comparative analyses of the two methods are not clearly described in the abstract.
Discussion and Conclusion refer to the new system developed by the author based on the model site historic analyses. The conclusion states that Watkins Glen Park could remain a popular scenic landscape destination for about 160 years in spite of the several development phases. The question is whether the balanced touristic interests are due to developments attached to and around the park site while the original natural like landscape remained sustainable. Unfortunately, it is only a question because no analyses are given on the condition of the most protected and valuable gorge landscape and ecosystem quality.
Round 2
Reviewer 5 Report
I appreciate very much all the improvements, especially the new figures, maps, and graphs. The paper is clear in this format with its new structure.